# A multi-omics framework for survival mediation analysis of high-dimensional proteogenomic data

Seungjun Ahn[1,2]*, Weijia Fu[1,2], Maaike van Gerwen[3], Lei Liu[4], Zhigang Li[5]

**1** Department of Population Health Science and Policy, Icahn School of Medicine at Mount Sinai, New York, New York, United States of America, **2** Tisch Cancer Center, Icahn School of Medicine at Mount Sinai, New York, New York, United States of America, **3** Department of Otolaryngology - Head and Neck Surgery, Icahn School of Medicine at Mount Sinai, New York, New York, United States of America, **4** Division of Biostatistics, Washington University in St. Louis, St. Louis, Missouri, United States of America, **5** Department of Biostatistics, University of Florida, Gainesville, Florida, United States of America

* seungjun.ahn@mountsinai.org

## Abstract

Survival analysis plays a crucial role in understanding time-to-event (survival) outcomes such as disease progression. Despite recent advancements in causal mediation frameworks for survival analysis, existing methods are typically based on Cox regression and primarily focus on a single exposure or individual omics layers, often overlooking multi-omics interplay. This limitation hinders the full potential of integrated biological insights. In this paper, we propose SMAHP, a novel method for survival mediation analysis that simultaneously handles high-dimensional exposures and mediators, integrates multi-omics data, and offers a robust statistical framework for identifying causal pathways on survival outcomes. This is one of the first attempts to introduce the accelerated failure time (AFT) model within a multi-omics causal mediation framework for survival outcomes. Through simulations across multiple scenarios, we demonstrate that SMAHP achieves high statistical power, while effectively controlling false discovery rate (FDR), compared with two other approaches. We further apply SMAHP to the largest head-and-neck carcinoma proteogenomic data, detecting a gene mediated by a protein that influences survival time. R package is freely available on CRAN repository and published under General Public License version 3.

## Author summary

In this study, we propose SMAHP, a novel multi-omics causal mediation framework that addresses the unique challenges of high-dimensional exposures, high-dimensional mediators, and survival outcomes. To our knowledge, this is the first methodological development specifically focused on survival causal mediation analysis in the context of multi-omics proteogenomic data. SMAHP incorporates a two-stage feature selection procedure combining penalization techniques and sure independence screening to efficiently identify relevant

**Data availability statement:** The pre-processed RNA-Seq, proteomics, and clinical data (metadata) from the National Cancer Institute-initiated CPTAC are available in the Proteomic Data Commons (https://pdc.cancer.gov/pdc/cptac-pancancer). The SMAHP (all upper cases) R package is freely available in the Comprehensive R Archive Network (CRAN) repository (https://cran.r-project.org/web/pack-ages/SMAHP/index.html).

**Funding:** The first author (S.A.) was supported in part by National Cancer Institute Cancer Center Support Grant P30CA196521 awarded to the Tisch Cancer Center of the Icahn School of Medicine at Mount Sinai. This work was supported by the Clinical and Translational Science Awards (CTSA) grant UL1TR004419 from the National Center for Advancing Translational Sciences. The funders had no role in study design, data collection and analysis, decision to publish, or preparation of the manuscript.

**Competing interests:** The authors have declared that no competing interests exist.

exposure and mediator candidates associated with survival outcomes. Through comprehensive simulation studies, we demonstrate the robustness of our approach. We further illustrate the practical utility of SMAHP by applying it to the largest proteogenomic dataset of head and neck cancer (CPTAC), uncovering a causal mediation pathway where a specific protein negatively mediates the effect of gene expression on survival time in patients with HPV-negative tumors. The proposed methodology is publicly available as an R package, SMAHP, on CRAN, accompanied by a detailed vignette to facilitate reproducibility and application.

## Introduction

Survival analysis is a powerful tool for understanding time-to-event outcomes, such as patient survival or disease progression. Recent advancements in high-throughput technologies have enabled the profiling of biological data on a scale that was once unimaginable, including proteomics data and RNA-Seq data. These advances open new avenues for uncovering complex relationships between biological entities and survival outcomes. For example, recent studies have identified protein or gene biomarkers associated with survival outcomes in patients with idiopathic pulmonary fibrosis [1], glioblastoma [2], hepatocellular carcinoma [3], small-cell lung carcinoma [4], cardiovascular diseases [5,8], Alzheimer's disease and related dementias [6], and oropharyngeal carcinoma [7], often using Cox proportional hazards (PH) regression models with or without penalization methods.

While traditional survival analysis methods, such as Cox PH regression, have been widely used to assess the direct effects of individual predictor variables on survival outcomes, they often fall short in capturing indirect effects or mediating pathways. Specifically, predictor models are fitted separately to assess associations between genes and survival, and between proteins and survival. The mediation analysis offers a solution by exploring how a hypothetical intermediate variable (i.e., mediator) bridges the effect of an exposure variable on an outcome. In the context of survival analysis, this approach provides valuable insights into how molecular biomarkers, such as protein expression levels, mediate the relationship between RNA-Seq gene expression and survival.

Over the past few years, pioneering methodological approaches have emerged for causal mediation frameworks with survival outcomes in the analysis of omics data. Notably, High-dimensional Mediation Analysis in Survival Models (HIMAsurvival) [42] first introduced a Cox PH regression framework, incorporating variable selection techniques via sure independence screening (SIS) [39] and minimax concave penalty (MCP)-penalization [37], to estimate and test the effects of high-dimensional mediators on survival outcome. Following this, another study presented a Cox regression-based framework [45], featuring adaptations of de-biased Lasso inference and a joint significance test for better false discovery control compared to HIMAsurvival [42]. More recently, the Mediation Analysis of Survival Outcomes and High-Dimensional Omics Mediators (MASH) [9] proposed a three-step high-dimensional mediator

selection procedure (i.e., pre-screening with SIS, MCP for variable selection, and FDR control using the Benjamini-Hochberg (BH) procedure). While the mediator selection procedure is similar to the two methods described earlier [42,45], MASH distinguishes itself by using $R^2$-based measures to estimate mediation effects, with its primary framework again centered on the Cox regression model.

Despite these advances in survival mediation analysis, two interconnected limitations persist. First, existing survival mediation approaches have primarily focused on high-dimensional mediators, with much less attention given to the setting of exposures. These studies typically consider a single binary or continuous exposure, overlooking the opportunity to capture the rich information present in high-dimensional exposure variables. A previous study [49] explored a frailty model (or Cox model with random effects) to identify mediation effects in the presence of high-dimensional exposures, but it only considers one mediator at a time. Second, these methods have showcased applications to individual omics layer (e.g., DNA methylation, metabolomics, and copy number variation data). However, biological systems are complex and driven by interplay across multiple omics layers. As demonstrated in numerous clinical and bioinformatics studies [10–13], the integration of multilayer (or multi-omics) analysis has been used to characterize key multi-omics pathways, offering a more comprehensive view of the molecular mechanisms underlying complex diseases such as cancer and Alzheimer's disease.

Significant gaps still remain, as only one study has attempted to address this issue in the context of survival causal mediation in the analysis of omics data [14]. This unified mediation analysis framework accounts for both multivariable exposures and mediators in relation to survival outcomes, applying it to proteogenomic data to identify genes mediated by proteins associated with survival. However, the primary focus was not specifically on survival outcomes. This study considered a Cox model, which may not be valid if the PH assumption is violated. Additionally, the simulations in the study did not explore a range of censoring rates (set to 50%).

The motivation of this paper is to develop a statistical methodology capable of (1) handling both high-dimensional mediators and exposures, (2) testing the mediation effect between exposure and survival outcomes with proper FDR control, and (3) analyzing multiple omics platforms simultaneously. In addition, from a clinical perspective, most existing mediation methods have primarily been developed for analyses within a single omics layer. In contrast, a growing body of clinical proteogenomic studies [15–21] has demonstrated the clinical relevance of jointly analyzing genomic and proteomic data, highlighting their complementary roles and associations with disease phenotypes across diverse disease types. Despite this clinical importance, high-dimensional causal mediation methods that explicitly integrate proteomic and transcriptomic data remain relatively underdeveloped.

To overcome these challenges and achieve the objectives of this study, we propose a novel approach to mediation analysis, specifically focused on survival outcomes and framed within the counterfactual paradigm. By implementing and applying this methodology to multilayered, high-dimensional proteogenomic data (high-dimensional genomic exposures from RNA-Seq data and high-dimensional protein mediators from proteomics data), we introduce this framework as Survival Mediation Analysis of High-dimensional Proteogenomic data (SMAHP). Our method procedure involves the following parts: First, we conduct a preliminary screening of high-dimensional mediators and exposures using penalized accelerated failure time (AFT) [35] and MCP-penalized regression models [37], based on their disjoint indirect effects (exposure→mediator and mediator→outcome) or direct effects (exposure→outcome). Second, using proteins and genes selected from the preliminary screening, we adopt the SIS [39] to identify "important" protein mediators that relate the gene exposures to the outcome. Third, a joint significance test [47] is performed to determine whether a particular mediator lies in the causal pathway between exposure and survival outcome, with appropriate control of the false discovery rate (FDR).

We demonstrate the advantages of SMAHP through comprehensive simulations and showcase its application to proteogenomic data from the Clinical Proteomic Tumor Analysis Consortium (CPTAC) to identify potential causal mediation pathways related to head and neck squamous cell carcinomas (HNSCC). We conclude with a discussion of the summary, challenges, limitations, and intended future directions for both application and methodological research.

## Results

### Simulation design

We generated exposure variables **X** from the multivariate normal distribution with no correlations between exposure variables. Each exposure variable has mean 0.4 and standard deviation 0.5. Two covariates **Z** = {$Z_1, Z_2$} were generated as follows: $Z_1 \sim N(0.12, 0.75^2), Z_2 \sim Bernoulli(0.3)$. Mediator variables **M** were then simulated from normal distribution with a proportion of **X** and **Z** associated with **M**. Specifically, 40% of the **M** are associated with 10% of **X** with an effect size of 0.8, as well as with $Z_1$ and $Z_2$ with effect sizes of 0.2 each, and a standard deviation of 0.5. Another 40% of the **M** are associated with 10% of the **X**, also with an effect size of 0.8, but with a standard deviation of 0.3. Additionally, 10% of the **M** are associated only with $Z_1$ and $Z_2$, with effect sizes of 0.2 and 0.3, respectively, and a standard deviation of 0.5. The remaining 10% of the **M** are not associated with any of the **X** or **Z**, with a standard deviation of 0.3. Survival outcomes $T$ were simulated using an AFT model, with $T$ associated with randomly selected **X** (effect size 0.8), **M** (effect size 4.0), and **Z** (effect size 0.12). The error term in the AFT model followed a normal distribution, and the censoring times were drawn from an exponential distribution, calibrated to achieve a 25% censoring rate. As part of sensitivity analyses, we also explored censoring rates of 50% and 75%. In comparison with the SMAHP, we compared performance using two different approaches. The first approach begins with a univariate marginal mediation and outcome model, using SIS to identify exposures, followed by a second SIS approach, described as Step 2, which we will refer to as SIS + SIS. The second approach is a naïve modeling method, where marginal mediation and outcome models are fitted without any penalization or the application of the SIS procedure. In addition, we conducted sensitivity analyses incorporating correlated gene and protein structures, considering correlations among exposure genes and among mediator proteins, with correlation levels set to 0.4. We also examined robustness by sampling the exposures from a negative binomial distribution with dispersion parameter 3 instead of a multivariate normal distribution. We included additional simulation experiments to assess the robustness of the proposed method to outliers. Specifically, 2% of the exposure values were generated as outliers by sampling from a uniform distribution over the interval [$\mu + 3\sigma, \mu + 3\sigma + 5$], where $\mu$ and $\sigma$ denote the mean and standard deviation of the normal distribution used to generate the exposures. We further evaluated performance using AFT models with two alternative residual distributions (Gamma and logistic error). For each setting, we repeated the simulation 200 times.

### Simulation results

In these simulation experiments, different combinations of $n$, $p$, and $k$ were considered. Table 1 summarizes the simulation results when the censoring rate is 25%. Across all scenarios, the naïve method significantly inflated the FDR, even though it achieved high power. In contrast, SMAHP maintained high power while adequately controlling the FDR at the 5% level. The size difference in power and FDR was small between SMAHP and SIS + SIS in Scenarios I and II. However, this difference was more pronounced in Scenarios III and IV. For instance, in Scenario III, our proposed method achieved a power of 0.8296, whereas SIS + SIS had a power of 0.6423 with a smaller sample size ($n = 200$). When the censoring rate was increased to 50% (see Table 2), the SIS + SIS approach no longer controlled the FDR, unlike our proposed method. This issue worsened as the censoring rate increased to 75% (see Table 3), regardless of the sample size adjustment. In all scenarios with such high 75%, SIS + SIS exhibited lower power and inflated FDR compared to SMAHP. On the other hand, SMAHP also had an inflated FDR with a smaller sample size, but it quickly regained control over FDR when the sample size was increased to $n = 400$. For instance, when the censoring rate was 75%, SMAHP achieved a power of 0.7618 and an FDR of 0.1082, which improved to a power of 0.9843 and an FDR of 0.0580 as the sample size increased.

As a whole, SMAHP required the most computation time compared with the SIS + SIS and naïve methods, as we recorded the average computational time. This is expected, as SMAHP leverages a penalization technique. See Tables 1 to 3. In addition, we explored different penalties for the mediation model in SMAHP, with the MCP-penalization used as

**Table 1. Simulation results using SMAHP (penalizations + SIS), SIS + SIS, and naïve approaches. The data were simulated with censoring rate of 25%.**

| Scenario | $p$ | $k$ | $n$ | Method | Power | FDR | Average Computational Time (in minutes) |
|---|---|---|---|---|---|---|---|
| I | 50 | 100 | 200 | Naïve | 0.9893 | 0.9442 | 0.32 |
| | | | | SIS + SIS | 0.9355 | 0.0370 | 0.11 |
| | | | | SMAHP | 0.9853 | 0.0313 | 0.98 |
| | | | 400 | Naïve | 0.9990 | 0.9456 | 0.38 |
| | | | | SIS + SIS | 0.9933 | 0.0316 | 0.12 |
| | | | | SMAHP | 1.0000 | 0.0337 | 2.26 |
| II | 50 | 200 | 200 | Naïve | 0.9908 | 0.9719 | 0.65 |
| | | | | SIS + SIS | 0.9203 | 0.0333 | 0.19 |
| | | | | SMAHP | 0.9780 | 0.0362 | 0.83 |
| | | | 400 | Naïve | 1.0000 | 0.9743 | 0.79 |
| | | | | SIS + SIS | 0.9968 | 0.0189 | 0.23 |
| | | | | SMAHP | 0.9995 | 0.0290 | 2.49 |
| III | 100 | 100 | 200 | Naïve | 0.9250 | 0.9464 | 0.51 |
| | | | | SIS + SIS | 0.6423 | 0.0215 | 0.16 |
| | | | | SMAHP | 0.8296 | 0.0114 | 1.13 |
| | | | 400 | Naïve | 0.9994 | 0.9457 | 0.60 |
| | | | | SIS + SIS | 0.9799 | 0.0081 | 0.21 |
| | | | | SMAHP | 0.9899 | 0.0244 | 2.50 |
| IV | 100 | 200 | 200 | Naïve | 0.9259 | 0.9731 | 1.02 |
| | | | | SIS + SIS | 0.6561 | 0.0238 | 0.31 |
| | | | | SMAHP | 0.8380 | 0.0192 | 1.76 |
| | | | 400 | Naïve | 0.9995 | 0.9739 | 1.44 |
| | | | | SIS + SIS | 0.9743 | 0.0144 | 0.41 |
| | | | | SMAHP | 0.9960 | 0.0205 | 3.97 |

Abbreviations: FDR, false discovery rate.

SMAHP = MCP penalized mediation model with a penalized AFT outcome model in Step 1, followed by SIS in Step 2; SIS + SIS = Univariate marginal mediation and outcome models, with SIS used to select exposures, followed by a similar SIS approach in Step 2; Naïve model = Marginal mediation and outcome models without any penalizations or SIS.

$n$ = sample size; $p$ = number of genes (exposures); $k$ = number of proteins (mediators).

the default. We investigated whether we would obtain similar or substantially different results with other penalties, such as the elastic-net and Lasso penalties. S1 Table summarizes this experiment, showing similar power and FDR, as well as comparable computational time. Additional sensitivity analyses, suggested by the reviewers, were conducted to further evaluate the performance of SMAHP. Specifically, we examined settings with correlated exposure genes and correlated mediator proteins (S2 Table). Across these settings, the results were broadly consistent with the primary simulation results, with SMAHP showing stable power and false discovery rate (FDR) control under low to moderate correlation levels. We also evaluated performance when exposures were sampled from a negative binomial distribution (S3 Table). In this setting, SMAHP showed similar behavior to the primary simulations, with satisfactory power and FDR control around the nominal 5% level. Sensitivity analyses incorporating outliers (S4 Table) indicated that SMAHP continued to perform well as the sample size increased. Finally, under two alternative residual distributions, Gamma (S5 Table) and logistic (S6 Table), results were consistent with the primary simulations, showing that the SMAHP maintained high power and

**Table 2. Simulation results using SMAHP (penalizations + SIS), SIS + SIS, and naïve approaches. The data were simulated with censoring rate of 50%.**

| Scenario | p | k | n | Method | Power | FDR | Average Computational Time (in minutes) |
|---|---|---|---|---|---|---|---|
| I | 50 | 100 | 200 | Naïve | 0.9828 | 0.9429 | 0.26 |
| | | | | SIS + SIS | 0.8680 | 0.0757 | 0.10 |
| | | | | SMAHP | 0.9603 | 0.0506 | 1.43 |
| | | | 400 | Naïve | 0.9983 | 0.9450 | 0.40 |
| | | | | SIS + SIS | 0.9865 | 0.0348 | 0.12 |
| | | | | SMAHP | 1.0000 | 0.0417 | 3.86 |
| II | 50 | 200 | 200 | Naïve | 0.9845 | 0.9712 | 0.51 |
| | | | | SIS + SIS | 0.8533 | 0.0919 | 0.19 |
| | | | | SMAHP | 0.9523 | 0.0581 | 1.14 |
| | | | 400 | Naïve | 1.0000 | 0.9742 | 0.59 |
| | | | | SIS + SIS | 0.9955 | 0.0423 | 0.22 |
| | | | | SMAHP | 0.9983 | 0.0448 | 3.85 |
| III | 100 | 100 | 200 | Naïve | 0.9228 | 0.9454 | 0.66 |
| | | | | SIS + SIS | 0.5096 | 0.0398 | 0.16 |
| | | | | SMAHP | 0.7678 | 0.0375 | 1.42 |
| | | | 400 | Naïve | 0.9993 | 0.9456 | 0.58 |
| | | | | SIS + SIS | 0.9484 | 0.0203 | 0.20 |
| | | | | SMAHP | 0.9843 | 0.0334 | 3.94 |
| IV | 100 | 200 | 200 | Naïve | 0.9234 | 0.9729 | 1.27 |
| | | | | SIS + SIS | 0.4836 | 0.0612 | 0.41 |
| | | | | SMAHP | 0.7270 | 0.0432 | 2.09 |
| | | | 400 | Naïve | 0.9995 | 0.9739 | 1.43 |
| | | | | SIS + SIS | 0.9419 | 0.0389 | 0.39 |
| | | | | SMAHP | 0.9776 | 0.0324 | 4.56 |

Abbreviations: FDR, false discovery rate.

SMAHP = MCP penalized mediation model with a penalized AFT outcome model in Step 1, followed by SIS in Step 2; SIS + SIS = Univariate marginal mediation and outcome models, with SIS used to select exposures, followed by a similar SIS approach in Step 2; Naïve model = Marginal mediation and outcome models without any penalizations or SIS.

$n$ = sample size; $p$ = number of genes (exposures); $k$ = number of proteins (mediators).

controlled the FDR. The entire set of simulations was run on the high-performance supercomputer Minerva at the Icahn School of Medicine at Mount Sinai, utilizing 10 CPU cores and 8 GB of RAM per node.

## Application study: Analysis of clinical proteomic tumor analysis consortium data

We are motivated by the problem of human papillomavirus-negative (HPV-Neg) HNSCC, which remains insufficiently studied, as highlighted by recent clinical research [17,53], despite HNSCC being ranked as the sixth most prevalent epithelial cancer globally [54]. Specifically, a recent CPTAC HNSCC study [17] has emphasized that a comprehensive understanding of how transcriptomic and molecular changes contribute to tumor phenotypes is still lacking, highlighting the critical need for further investigation in this area.

In this study, pre-processed RNA-Seq, proteomics, and clinical data (metadata) from the National Cancer Institute-initiated CPTAC were downloaded from the Proteomic Data Commons (https://pdc.cancer.gov/pdc/cptac-pancan-cer), which is one of the largest public repositories of proteogenomic data. The CPTAC has been utilized to increase

**Table 3. Simulation results using SMAHP (penalizations + SIS), SIS + SIS, and naïve approaches. The data were simulated with censoring rate of 75%.**

| Scenario | p | k | n | Method | Power | FDR | Average Computational Time (in minutes) |
|---|---|---|---|---|---|---|---|
| I | 50 | 100 | 200 | Naïve | 0.9635 | 0.9404 | 0.32 |
| | | | | SIS + SIS | 0.4148 | 0.4595 | 0.11 |
| | | | | SMAHP | 0.8165 | 0.1058 | 0.98 |
| | | | 400 | Naïve | 0.9920 | 0.9435 | 0.38 |
| | | | | SIS + SIS | 0.9415 | 0.1604 | 0.12 |
| | | | | SMAHP | 0.9845 | 0.0532 | 2.26 |
| II | 50 | 200 | 200 | Naïve | 0.9585 | 0.9697 | 0.65 |
| | | | | SIS + SIS | 0.3433 | 0.5680 | 0.19 |
| | | | | SMAHP | 0.7618 | 0.1082 | 0.83 |
| | | | 400 | Naïve | 0.9963 | 0.9730 | 0.79 |
| | | | | SIS + SIS | 0.9105 | 0.1658 | 0.23 |
| | | | | SMAHP | 0.9843 | 0.0580 | 2.49 |
| III | 100 | 100 | 200 | Naïve | 0.9076 | 0.9436 | 0.51 |
| | | | | SIS + SIS | 0.2140 | 0.4940 | 0.16 |
| | | | | SMAHP | 0.4559 | 0.2082 | 1.13 |
| | | | 400 | Naïve | 0.9985 | 0.9451 | 0.60 |
| | | | | SIS + SIS | 0.7844 | 0.1598 | 0.21 |
| | | | | SMAHP | 0.9466 | 0.0580 | 2.50 |
| IV | 100 | 200 | 200 | Naïve | 0.9095 | 0.9718 | 1.02 |
| | | | | SIS + SIS | 0.1599 | 0.6020 | 0.31 |
| | | | | SMAHP | 0.3819 | 0.2813 | 1.76 |
| | | | 400 | Naïve | 0.9984 | 0.9734 | 1.44 |
| | | | | SIS + SIS | 0.7044 | 0.2317 | 0.41 |
| | | | | SMAHP | 0.9099 | 0.0549 | 3.97 |

Abbreviations: FDR, false discovery rate.

SMAHP = MCP penalized mediation model with a penalized AFT outcome model in Step 1, followed by SIS in Step 2; SIS + SIS = Univariate marginal mediation and outcome models, with SIS used to select exposures, followed by a similar SIS approach in Step 2; Naïve model = Marginal mediation and outcome models without any penalizations or SIS.

$n$ = sample size; $p$ = number of genes (exposures); $k$ = number of proteins (mediators).

understanding of the molecular mechanisms of cancer through its large-scale, mass spectrometry-based proteomic profiling data of tumor samples, which were previously analyzed by the Cancer Genome Atlas (TCGA) [52]. In comparison to TCGA, CPTAC provides a more extensive proteome coverage.

The overall survival (OS) is defined as the time from cancer diagnosis to death. Patients were censored if the event had not occurred by the last time of follow-up. The median OS is 46.27 months, and the Kaplan-Meier curve is presented in S1 Fig. The original sample size consisted of 109 patients, with 60,669 genes and 9,469 proteins available in the RNA-Seq and proteomics data, respectively. Prior to the application of the SMAHP, a univariate AFT model was fitted to each gene and protein separately for pre-screening purposes in this ultra high-dimensional data. The top 100 genes and top 200 proteins with the smallest p-values were then selected for analysis using SMAHP. Seven patients did not have OS data available and were excluded, resulting in a final sample size of 102 patients, with a censoring rate of 67.6%. After applying SMAHP, there was indirect effect ($p$ = 0.001) of late cornified envelope 3E protein (LCE3E; Ensembl ID: ENSG00000185966.4) on the association between high-mobility group box 1 pseudogene 23 (HMGB1P23; Ensembl

ID: ENSG00000253770.1) and OS, with age included as an additional covariate. Fig 1 provides a summary of the analysis. While the direct effect on the survival time is positive, it appears that the indirect effect through the mediator is negative. Such a pattern may arise when the exposure influences survival through multiple pathways, with the mediator capturing an effect in the opposite direction to other pathways such as the direct pathway. This observation highlights the complexity of the underlying biological processes and is provided to aid interpretation of the estimated effects.

Although members of the high-mobility group box family have been implicated in carcinogenesis (e.g., high expression of HMGB1 in human nasopharyngeal carcinoma) and autoinflammatory diseases [55], no published literature exists on diseases or disorders specifically associated with HMGB1P23. A search of the MalaCards human disease database (https://www.malacards.org/) [56] also did not yield relevant findings. Interestingly, the MalaCards revealed that LCE3E is associated with plantar warts, which are caused by HPV. However, given that the study samples tested HPV-negative, this association should be interpreted with caution. Reported LCE3E-related pathways include keratinization and nervous system development. The total runtime to complete the analysis using SMAHP was 2.28 minutes. The analysis was performed on a 2023 MacBook Pro equipped with an M3 processor and 16GB of RAM.

## Discussion

In this study, we introduce SMAHP, a novel multi-omics causal mediation framework designed to handle high-dimensional exposures, high-dimensional mediators, and survival outcomes. To the best of our knowledge, this is the first framework in the literature focused specifically on identifying causal mediation pathways for time-to-event outcomes using multi-omics data. SMAHP has several key features that make it particularly useful in practical applications where existing methods are not feasible. First, it identifies causal pathways more accurately and tests the validity of the indirect effects within these pathways. Unlike other methods, which often consider a single binary exposure or a single continuous exposure at a time when assessing causal pathways, SMAHP is designed to handle multiple exposures and mediators simultaneously. Second, SMAHP enables researchers to gain a more comprehensive understanding of biological and molecular mechanisms by mapping gene-protein-outcome pathways, rather than analyzing gene-outcome and protein-outcome relationships separately. This is particularly valuable, as highlighted in the introduction, where emerging research underscores the importance of the synergy between proteomics and genomics and their connections to phenotypes across various disease types through proteogenomic analysis [15–21]. Third, the outcome model in our method is based on the AFT model, which is not constrained by the PH assumption, a key requirement of the Cox model. In contrast, all other existing methods rely on the Cox model.

Our simulation study demonstrated that, at least for the scenarios considered, SMAHP maintains high statistical power while appropriately controlling FDR. In general, this pattern persists and even outperforms the SIS + SIS and naïve approaches across different censoring rates, although a larger sample size is needed to restore high power and proper FDR control when the censoring rate is very high (i.e., 75%). These findings highlight the importance of the censoring rate when applying the proposed method and motivate further discussion of its implications for study design. Censoring rate

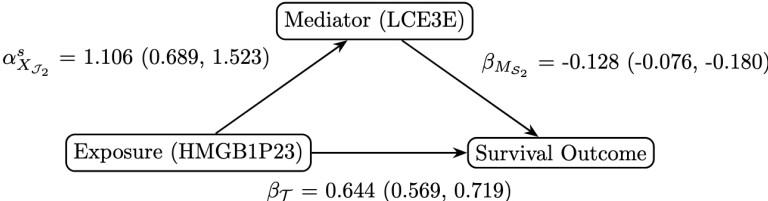

**Fig 1. A directed acyclic graph summarizing the analysis of CPTAC HNSCC data, highlighting direct and indirect effects with 95% confidence intervals in parentheses using SMAHP.**

is an important consideration when applying the proposed method and has direct implications for study design. Higher censoring rates generally reduce the effective information available for inference, which can lead to decreased statistical power. Our simulation results demonstrate that under high censoring conditions, such as 75%, increasing the sample size can substantially improve power while maintaining appropriate control of the false discovery rate. From a practical perspective, when researchers anticipate a high censoring rate in a biological dataset, larger sample sizes may be required to achieve adequate performance. In contrast, for studies with relatively low censoring rates, the method performs well even with more moderate sample sizes. Therefore, understanding the expected censoring mechanism and rate in a given study can inform decisions regarding sample size planning and the feasibility of applying the proposed framework. In our real-data application to the CPTAC HNSCC dataset, we identified a significant mediating effect of LCE3E on the association between the relatively unexplored HMGB1P23 gene and survival time among HPV-Neg study population. At present, the proposed method has been demonstrated using a single multi-omics, high-dimensional dataset, and an independent dataset was not available to serve as a traditional validation set. As a result, the same causal pathway may or may not be observed across different datasets, and differences in findings could reflect variations in cohort characteristics, such as age or ethnicity, rather than limitations of the proposed methodology. In future studies, we will try to validate the proposed approach using independent multi-omics datasets with larger sample sizes (if available).

There are several areas where further work is needed, and future extensions are possible. In particular, we explored alternative multiple hypothesis testing controls beyond the BH procedure, including the q-value method [51], Westfall-Young correction [60], and HDMT [59]. However, these approaches were not adopted, as they either resulted in worse performance or were incompatible with modeling high-dimensional exposures (the latter being the case for HDMT). Future methodological development that better accommodates high-dimensional exposure settings could make it possible to incorporate these alternative multiple testing approaches. Given the interdependencies between biological entities, incorporating group-level biological information, such as biological pathways or protein complexes, could be a crucial next step in advancing the analysis [57,58]. By grouping genes (and proteins) based on existing biological knowledge, this approach would help identify how series of interconnected molecular interactions work together to specific biological processes, which we loosely describe as a "pathway-level mediation framework". One potential way to achieve this is to leverage multilevel or generalized linear mixed models to account for clustered data.

Additionally, we considered the penAFT method [35] for the penalized outcome model. We believe that penalization methods for the AFT model are less studied in the literature compared to those for the Cox model, likely because the latter offers the advantage of estimating hazard ratios, which are commonly used to compare biological conditions. Future research is needed to develop more computationally and statistically robust penalization algorithms to enhance the identification of exposures and mediators in the AFT outcome model. For example, in our study, we assumed a parametric AFT model, with the error term following a normal distribution. In future studies, it would be interesting to consider a nonparametric AFT model. This approach would eliminate the need for pre-processed, normalized data, such as the bioinformatics normalization used to align raw data from different samples, and would help reduce technical noise by not relying on parametric assumptions. Future work may consider extensions incorporating non-linear or interaction effects. Although feasible in principle, such extensions would substantially increase computational burden in high-dimensional settings, and interactions between continuous variables may be difficult to interpret biologically, potentially limiting their practical utility. From a computational perspective, further improvements may also be achieved by leveraging parallel computing strategies and incorporating dimensionality reduction or feature screening steps with marginal screening to reduce computational complexity when analyzing increasingly high-dimensional multi-omics data. Finally, a typical limitation of high dimensional mediation modeling methods is that it cannot disentangle the potential sequential causal mediation effect through multiple mediators due to the complexity induced by the high dimensionality of mediators. As with many existing methods, our proposed method does not test any sequential causal mediation effects or the cumulative causal mediation effect of multiple mediators.

## Materials and methods

### Notations and assumptions

Herein, we consider a proteogenomic data with a survival outcome $T$, a vector of $\boldsymbol{M} = \{M_1, \ldots, M_K\}$ proteomes as mediators, a vector of $\boldsymbol{X} = \{X_1, \ldots, X_P\}$ genes as exposures, and additional covariates $\boldsymbol{Z} = \{Z_1, \ldots Z_Q\}$ to be adjusted for such as age, sex, smoking history, alcohol consumption categories, immune score, and histologic grades from $n$ i.i.d. observations. For better readability, the subject-level subscripts are suppressed unless otherwise stated.

In this paper, our causal mediation model is implemented within the counterfactual framework [22,23]. In counterfactual notations, we will let $M_k(x)$ denote the potential outcome of the $k$th proteome when each $\boldsymbol{X}$ is set to $x$, and let $T_{x,M_k(x)}$ be the potential outcome of $T$ with an observed expression levels for all genes when $\boldsymbol{X} = x$.

The causal effects are assessed by taking the mean expected difference in counterfactual outcomes that would have been observed [24,25]. In the log-scale, we define the interventional indirect (IIE) and interventional direct effect [61,62] with any two levels of continuous exposure by the decomposition of a total effect [26].

$$\text{IIE} = E[\log(T_{x^*,G_k(x^*)}) - \log(T_{x^*,G_k(x)})],$$
$$\text{IDE} = E[\log(T_{x^*,G_k(x)}) - \log(T_{x,G_k(x)})],$$

where $x^* \neq x$ for all genes in $\boldsymbol{X}$ and $G_k(x)$ or $G_k(x^*)$ is a random draw from the distribution of $M_k(x)$ or $M_k(x^*)$. The exposure values are rescaled in order to assess the changes in causal quantities after the unit increase in original exposure value. The estimates of IDE and IIE are obtained from the mediation and outcome models, respectively, which are discussed in Model Specifications subsection.

We will in addition assume that the IDE and IIE are estimated under the assumptions of no-unmeasured confounders [27–29] for the exposure-outcome (Eq 1), mediator-outcome (Eq 2), and exposure-mediator (Eq 3). That is, for each $p = 1, \ldots, P$ exposure,

$$X_p \perp T_{xM(x)} \mid \boldsymbol{Z}, \tag{1}$$

$$M(x) \perp T_{xM(x)} \mid \boldsymbol{X}, \boldsymbol{Z}, \tag{2}$$

$$\boldsymbol{X} \perp M(x) \mid \boldsymbol{Z}. \tag{3}$$

### Model specifications

We consider the following models to describe the causal relationships illustrated in Fig 2. For the outcome model (Eq 4), the high-dimensional accelerated failure time (AFT) model [30–32] is used to estimate and test the effect of mediator $\boldsymbol{M}$ (proteomes) in the causal pathway between continuous exposures $\boldsymbol{X}$ (gene expressions) on the survival outcome $T$, while accounting for clinically meaningful covariates $\boldsymbol{Z}$. The data we use to fit this outcome model also includes $\delta$, the censoring indicator for the log censoring time $\log(T)$. Furthermore, without loss of generality, the features are centered to eliminate an intercept. For the mediator model (Eq 5), the linear regression is fitted to model the association between $\boldsymbol{X}$ and each mediator $M_k$ for $k = 1, \ldots, K$.

$$\log(T \mid \boldsymbol{M}, \boldsymbol{Z}, \boldsymbol{X}) = \beta_{\boldsymbol{X}}^\top \boldsymbol{X} + \beta_{\boldsymbol{Z}}^\top \boldsymbol{Z} + \beta_{\boldsymbol{M}}^\top \boldsymbol{M} + b\varepsilon, \tag{4}$$

$$M_k = \alpha_{\boldsymbol{X}}^{k\top} \boldsymbol{X} + \alpha_{\boldsymbol{Z}}^{k\top} \boldsymbol{Z} + \xi_k, \tag{5}$$

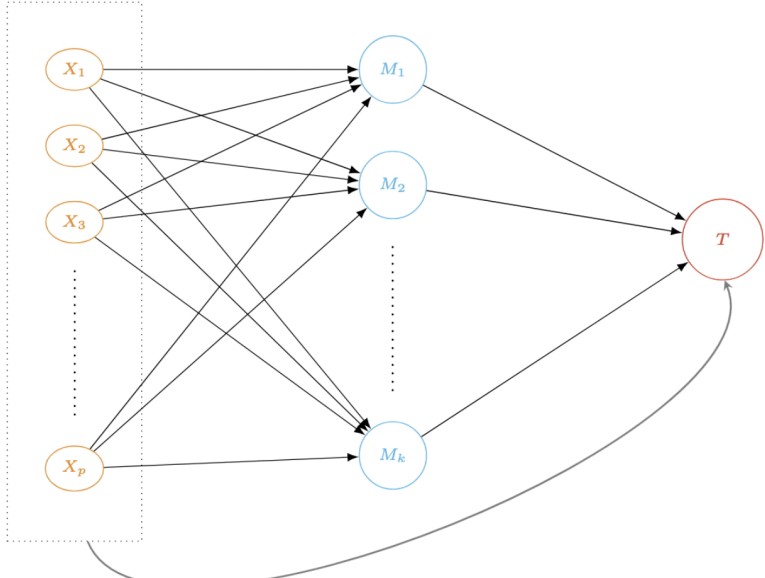

**Fig 2. A causal diagram to describe the framework of the mediation analysis of high-dimensional mediators with multivariable exposures and survival endpoint.** Here, the two types of effects are assessed: (1) the global indirect effect $\alpha_X^{k\top}\beta_M$ between $\boldsymbol{X}$ and $T$, and specific mediation effect that is mediated by a mediator variable $M_k$ for each $k = 1, \ldots, K$ proteome that is selected from the penalized variable selection (nodes colored in blue) and (2) the direct effect $\beta_X^\top$ between multivariable exposures $\boldsymbol{X} = X_1, \ldots, X_p$ RNA-Seq gene expressions (nodes colored in orange) and survival outcome $T$ (a node colored in red).

where $\beta_X^\top = (\beta_{X_1}, \ldots, \beta_{X_P})$ is the regression coefficients vector for the effect of $\boldsymbol{X}$ on $T$; $\beta_Z^\top = (\beta_{Z_1}, \ldots, \beta_{Z_Q})$ is a vector of regression coefficients for the covariates; $\beta_M^\top = (\beta_{M_1}, \ldots, \beta_{M_K})$ is the regression coefficients vector of the effect of $\boldsymbol{M}$ on $T$ with the presence of $\boldsymbol{X}$; and $\epsilon$ is a random error variable following the log-Weibull distribtution (flexible such as log-normal and other distributions) with the scale parameter $b$ [33]. $\alpha_X^{k\top} = (\alpha_{X_1}^k, \ldots, \alpha_{X_P}^k)$ is a vector of regression coefficients for the effect of $\boldsymbol{X}$ on each $M_k$; $\alpha_Z^{k\top} = (\alpha_{Z_1}^k, \ldots, \alpha_{Z_Q}^k)$ is the regression coefficients vector for covariates; and $\xi_k$ is normal random error. It is important to note that estimates of $\beta_X^\top$ lead to the IDE of $\boldsymbol{X}$ on $T$ (i.e., $\boldsymbol{X} \to T$). The IIE of $M_k$ on the causal pathway between $\boldsymbol{X}$ and $T$ (i.e., $\boldsymbol{X} \to M_k \to T$) is defined by the product rule of estimates $\alpha_X^k \beta_M^\top$ [32,34,45].

### Step 1: Penalization for selecting mediator and exposure candidates

**Penalization for outcome model.** It is likely that proteomes are highly correlated with one another, as are genes. Moreover, it is essential to assess the associations between these biological entities and the outcome. Therefore, we consider penalized regression technique to identify proteomic mediators and genomic exposures, following the recent work [35] on the penalized AFT model using a variation of the alternating direction method of multipliers algorithm. For the $i$th subject, we can derive the following equations from the outcome model (Eq 4)

$$\log t_i = \boldsymbol{X}^\top \beta_X + a\varepsilon_i,$$
$$\log t_i = \boldsymbol{M}^\top \beta_M + b\varepsilon_i, \quad \text{for } i = 1, \ldots, n,$$

where the outcome model penalization is applied separately for $\boldsymbol{X}$ and $\boldsymbol{M}$, respectively. The estimated active sets, denoted as $\mathcal{S}_1$ and $\mathcal{T}$, will be determined from each of these models. $\mathcal{S}_1$ refers to the active set of pre-screened significant genes

for $\boldsymbol{X} \to T$, and $\mathcal{T}$ refers to the active set of pre-screened significant proteins for $\boldsymbol{M} \to T$. For more details on the penalized AFT regression, refer to the S1 Appendix.

**Penalization for mediation model.** We identify important mediators and exposures in the outcome model through the penalization approach detailed in Penalization for Outcome Model subsection earlier. Concurrently, we consider penalized mediation models for each mediator, minimizing the penalties based on the minimax concave penalty (MCP) approach [37] for the marginal mediation model (Eq 5). For $i$th subject, we express the marginal model for mediators as follows

$$M_{ki} = \boldsymbol{X}_i^\top \alpha_{\boldsymbol{X}}^k + \xi_{ki}, \text{ for } i = 1, \ldots, n.$$

See the S1 Appendix for more details on MCP estimates. From this penalized mediation model, the estimated active set $\mathcal{J}_1^k$ will be identified, consisting of pre-screened significant genes for $\boldsymbol{X} \to M_k$. In our simulation study, we compared its performance with that of elastic-net [38] and $L_2$ regularization (Lasso) [36] when modeling the mediator marginally.

Furthermore, we can rewrite the outcome and mediation models derived from these penalization steps as follows:

$$\log(T \mid \boldsymbol{M}_{\mathcal{S}_1}) = \beta_{\boldsymbol{M}_{\mathcal{S}_1}}^\top \boldsymbol{M}_{\mathcal{S}_1} + b\varepsilon, \tag{6}$$

$$\log(T \mid \boldsymbol{X}_\mathcal{T}) = \beta_{\boldsymbol{X}_\mathcal{T}}^\top \boldsymbol{X}_\mathcal{T} + b\varepsilon, \tag{7}$$

$$M_k = \alpha_{\boldsymbol{X}_{\mathcal{J}_1^k}}^{k\top} \boldsymbol{X}_{\mathcal{J}_1^k} + \xi, k \in \mathcal{S}_1 \tag{8}$$

where the focus is shifted to (i) $\mathcal{S}_1$ being the estimated active set (*i.e.,* non-zero coefficients) of $K$ mediators selected as candidate proteomes to be extensively studied based on the penalized outcome model in Eq 6; (ii) $\mathcal{T}$ is the estimated active set of $P$ exposures selected as candidate genes from the penalized outcome model in Eq 7; and lastly, (iii) $\mathcal{J}_1^k$ is the estimated active set of $P$ exposures selected as candidate genes from the penalized mediation model in Eq 8.

## Step 2: Screening important mediators with control for exposures

In Step 1, candidate proteomes and genes are selected from the penalized AFT and MCP regression models. However, these candidates are chosen based on either the disjoint indirect effect (i.e., $\boldsymbol{X} \to M_k$ and $\boldsymbol{M} \to T$, separately) or the direct effect (i.e., $\boldsymbol{X} \to T$). Hence, in Step 2, we reformulate the models defined in Eqs 6–8 into those specified for the causal mediation framework in Eqs 4 and 5. The objective of this section is to identify "important" proteomic mediators that account for exposures and to further screen potential mediators in order to reduce dimensionality in high-dimensional data, thereby boosting computational efficiency.

For each $s \in \mathcal{S}_1$, we consider the outcome and mediation models as redefined

$$\log(T \mid M_s, \boldsymbol{X}_\mathcal{T}) = \beta_{\boldsymbol{X}_\mathcal{T}}^\top \boldsymbol{X}_\mathcal{T} + \beta_{M_s} M_s + b\varepsilon,$$
$$M_s = \alpha_{\boldsymbol{X}_{\mathcal{J}_1^s}}^{s\top} \boldsymbol{X}_{\mathcal{J}_1^s} + \xi,$$

where $\beta_{\boldsymbol{X}_\mathcal{T}}$ can be estimated by the maximum likelihood estimators of the AFT outcome model, and $\alpha_{\boldsymbol{X}_{\mathcal{J}_1^s}}^s$ can be estimated by the ordinary least squares estimators of the linear regression mediation model, respectively.

Leveraging the sure independence screening (SIS) approach [39], the pairs $(M_s, \boldsymbol{X}_{\mathcal{J}_1^s})$ among the top $n/\log(n)$ largest values of $|\alpha_{\boldsymbol{X}_{\mathcal{J}_1^s}}^s \cdot \beta_{M_s}|$ are screened. If a pair exhibits meaningful mediation effects, the mediators and genes are selected and denoted as $\mathcal{S}_2$ and $\mathcal{J}_2^s$, where $\mathcal{S}_2$ is a subset of $\mathcal{S}_1$, and $\mathcal{J}_2^s$ is the subset of $\mathcal{J}_1^s$ for any $s \in \mathcal{S}_2$, respectively. Otherwise, pairs without replacing "with minimal" mediation effects are dropped. SIS is a computationally efficient method that quickly reduces the dimensionality while retaining the variables in the model with higher correlations [40]. The threshold

for selecting pairs are typically $n/\log(n)$ and $n-1$, as used without formal theoretical justificaiton in the original SIS methodology paper [39]. However, the threshold can be increased to a multiple of $n/\log(n)$, such as $2n/\log(n)$ or $3n/\log(n)$, to increase the probability of identifying important mediators, as demonstrated in other studies [41–44].

### Step 3: Hypothesis Testing

In the earlier subsections, penalized regression approaches are used for variable selection (Step 1) and SIS for identifying "important" mediators (Step 2). By incorporating these mediators, exposures, and clinically meaningful covariates, we can express the outcome and mediation models as

$$\log(T \mid \boldsymbol{M}_{\mathcal{S}_2}, \boldsymbol{Z}, \boldsymbol{X}_{\mathcal{T}}) = \beta_{\mathcal{T}}^{\top}\boldsymbol{X}_{\mathcal{T}} + \beta_{\boldsymbol{Z}}^{\top}\boldsymbol{Z} + \beta_{\boldsymbol{M}_{\mathcal{S}_2}}^{\top}\boldsymbol{M}_{\mathcal{S}_2} + b\varepsilon, \tag{9}$$

$$M_s = \alpha_{\boldsymbol{X}_{\mathcal{J}_2^s}}^{s\top}\boldsymbol{X}_{\mathcal{J}_2^s} + \alpha_{\boldsymbol{Z}}^{s\top}\boldsymbol{Z} + \xi_s \text{ for each } s \in \mathcal{S}_2 \tag{10}$$

where $\mathcal{S}_2$ and $\mathcal{J}_2^s$ are the active sets of "important" mediators and exposures identified in Step 2, respectively. The coefficient $\beta_{\boldsymbol{M}_{\mathcal{S}_2}}$ corresponds to estimates for the mediators within $\mathcal{S}_2$. We have identified exposures that impose greater causal effect when paired with $M_s$ derived from the SIS. These exposures are subsequently regressed in the mediation model as described in Eq 10.

The joint significance (JS) test [47] is adopted to test whether a particular proteomic mediator lies in the causal pathway from an genomic exposure to a survival outcome (i.e., $H_{0,s} : \alpha_{\boldsymbol{X}_{\mathcal{J}_2^s}}^s \beta_{\boldsymbol{M}_{\mathcal{S}_2}} = \boldsymbol{0}$ vs. $H_{1,s} : \alpha_{\boldsymbol{X}_{\mathcal{J}_2^s}}^s \beta_{\boldsymbol{M}_{\mathcal{S}_2}} \neq \boldsymbol{0}$ for $s = 1, \ldots, r$, where $r = |\mathcal{S}_2|$ denotes the cardinality of a set $\mathcal{S}_2$). The JS test (also referred to as the JS-uniform test) has been utilized in several studies [42,46] and is recognized for its ability to control the type I error rate while maintaining statistical power [48]. The primary distinction between our study and the aforementioned studies [42,46] lies in the number of exposures being tested for each mediator. Accordingly, for $j = 1, \ldots, u$, where $u = |\mathcal{J}_2|$ genes, we can articulate the null and research hypotheses as follows

$$H_{0,sj} : \alpha_{X_j}^s \beta_{M_s} = 0 \text{ vs. } H_{1,sj} : \alpha_{X_j}^s \beta_{M_s} \neq 0. \tag{11}$$

to determine the gene-specific (or elementwise) indirect effect for each proteome mediator. The null hypothesis in Eq 11 can be further decomposed to three disjoint null sub-hypotheses

$$H_{0,sj} = \begin{cases} H_{10,sj} : \alpha_{X_j}^s \neq 0 \text{ and } \beta_{M_s} = 0, \\ H_{01,sj} : \alpha_{X_j}^s = 0 \text{ and } \beta_{M_s} \neq 0, \\ H_{00,sj} : \alpha_{X_j}^s = 0 \text{ and } \beta_{M_s} = 0. \end{cases}$$

Let $\alpha_{\boldsymbol{X}_{\mathcal{J}_2}}^s$ denote a $u \times r$ matrix of the mediation-exposure associations from $r$ mediation models, $\beta_{\boldsymbol{M}_{\mathcal{S}_2}}$ be a $r$-vector of mediator-outcome associations from the penalized outcome model, and $P_{\max}$ be a $u \times r$ matrix containing elementwise p-values. The elementwise p-value $P_{\max_{j,s}}$ for testing hypotheses in Eq 11 can be obtained by (i) comparing the p-values from the $s$th row of $\alpha_{\boldsymbol{X}_{\mathcal{J}_2}}^s$ with that from $\beta_{\boldsymbol{M}_{\mathcal{S}_2}}$ and (ii) taking the maximum of the two p-values.

$$P_{\max_{j,s}} = \max(P_{\alpha_{X_j}^s}, P_{\beta_{M_s}}), \tag{12}$$

where $P_{\alpha_{X_j}^s}$ and $P_{\beta_{M_s}}$ are defined as

$$P_{\alpha_{X_j}^s} = 2\left\{1 - F\left(\frac{|\hat{\alpha}_{X_j}^s|}{\hat{\sigma}_{\alpha_{X_j}^s}}\right)\right\},$$

$$P_{\beta_{M_s}} = 2\left\{1 - F\left(\frac{|\hat{\beta}_{M_s}|}{\hat{\sigma}_{\beta_{M_s}}}\right)\right\},$$

where $\hat{\beta}_{M_s}$ and $\hat{\alpha}^s_{X_j}$ are regression coefficient estimates from fitted models using Eqs 9 and 10, respectively. $\hat{\sigma}_{\beta_{M_s}}$ and $\hat{\sigma}_{\alpha^s_{X_j}}$ are estimates of standard error for $\hat{\beta}_{M_s}$ and $\hat{\alpha}^s_{X_j}$. $F(\cdot)$ is a standard normal cumulative distribution.

It is important to appropriately control the false discovery rate (FDR) for multiple hypothesis testing. Therefore, the BH-adjusted p-value [50] is applied on the $P_{\text{max}}$ matrix using `stats R` package.

## Supporting information

**S1 Appendix. Penalized outcome and mediation models in Step 1.** Detailed derivation of the penalized AFT model and MCP-penalized mediation model used for screening in Step 1.
(PDF)

**S1 Fig. A Kaplan-Meier curve for CPTAC-HNSCC application study.** Kaplan-Meier survival curve of overall survival for HPV-negative patients with head and neck squamous cell carcinoma from the CPTAC HNSCC dataset.
(TIF)

**S1 Table. Simulation results of the SMAHP with varying penal- ties in the mediation model.** The default MCP penalized mediation model is compared with mediation models using elastic-net and Lasso penalties.
(PDF)

**S2 Table. Simulation results for the SMAHP model (penalization + SIS) with correlated gene and protein structures.** SMAHP was assessed with correlated gene and protein structures.
(PDF)

**S3 Table. Simulation results of the SMAHP where exposures were generated from a negative binomial distribution, with a censoring rate of 25%.** SMAHP was further evaluated under this alternative exposure distribution.
(PDF)

**S4 Table. Simulation results of the SMAHP in the presence of outliers, with censoring rates of 25%.** We further evaluated the performance of SMAHP under this setting with outliers.
(PDF)

**S5 Table. Simulation results of SMAHP under a Gamma error distribution with censoring rates of 25%.** SMAHP was evaluated under Gamma residual distributions.
(PDF)

**S6 Table. Simulation results of SMAHP under a logistic error distribution with censoring rates of 25%.** SMAHP was evaluated under logistic residual distributions.
(PDF)

## Acknowledgments

We gratefully acknowledge the Minerva high-performance computing system, provided by Scientific Computing and Data at the Icahn School of Medicine at Mount Sinai. We would also like to thank Dr. Scott Roof (Icahn School of Medicine at Mount Sinai) for his valuable review and support of this manuscript.

## Author contributions

**Conceptualization:** Seungjun Ahn, Zhigang Li.

**Data curation:** Seungjun Ahn.

**Formal analysis:** Seungjun Ahn, Weijia Fu.

**Funding acquisition:** Maaike van Gerwen.

**Investigation:** Seungjun Ahn, Zhigang Li.

**Methodology:** Seungjun Ahn, Lei Liu, Zhigang Li.

**Project administration:** Seungjun Ahn.

**Resources:** Maaike van Gerwen.

**Software:** Seungjun Ahn, Weijia Fu.

**Supervision:** Seungjun Ahn.

**Validation:** Seungjun Ahn, Weijia Fu, Zhigang Li.

**Visualization:** Seungjun Ahn.

**Writing – original draft:** Seungjun Ahn.

**Writing – review & editing:** Seungjun Ahn, Weijia Fu, Maaike van Gerwen, Lei Liu, Zhigang Li.

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
