## [Decision Letter · Decision Letter 0]

12 Nov 2025

PCOMPBIOL-D-25-01410

A multi-omics framework for survival mediation analysis of high-dimensional proteogenomic data

PLOS Computational Biology

Dear Dr. Ahn,

Thank you for submitting your manuscript to PLOS Computational Biology. After careful consideration, we feel that it has merit but does not fully meet PLOS Computational Biology's publication criteria as it currently stands. Therefore, we invite you to submit a revised version of the manuscript that addresses the points raised during the review process.

We look forward to receiving your revised manuscript.

Kind regards,

Chris Amos

Guest Editor

PLOS Computational Biology

Ferhat Ay

Section Editor

PLOS Computational Biology

**Additional Editor Comments:**

This manuscript proposes a novel approach to mediation analysis for survival data and provides a very informative approach that uses an accelerated failure time (AFT) model for the analysis. For survival analysis of observational data from cancer studies, the AFT model is preferable to the usual Cox Proportional Hazards model because it does not make a proportional hazards assumption, which will be violated if there are substantive effects of a genetic factor on survival times. The manuscript is novel and important because it provides key theoretical developments for the application of AFT models in the study of expression data. That said, the reviewers all suggested major revisions. The key issue for these reviewers has been the assumption that the expression levels that are studied are normally distributed and uncorrelated with each other, with the lack of correlation being the more significant concern. In fact, most expression data shows clusters of highly correlated genes and the simulations need to at least evaluate what happens in the presence of some correlations and better would be to simulate clusters of correlated genes and evaluate the impact that will have on the analysis. Also, in the actual application to CPTAC only a single gene-mediator example is given. Since this is a multiomic method a much stronger and more illustrative example would be provided if the authors would evaluate more than one expression level jointly. The reviewers had other concerns about the method that should be addressed. All of them thought a major revision is needed but the significance and importance of this new method makes this an important contribution, if their points can be addressed. I apologize for delays in getting the review back but it was difficult finding reviewers for this important but complex contribution.

**Journal Requirements:**

3) Please amend your detailed Financial Disclosure statement. This is published with the article. It must therefore be completed in full sentences and contain the exact wording you wish to be published.

2) If any authors received a salary from any of your funders, please state which authors and which funders..

**Reviewers' comments:**

Reviewer's Responses to Questions

**Comments to the Authors:**

Reviewer #1: Review uploaded as attachment.

Reviewer #2: This study presents a multi-omics survival mediation framework of significant methodological value, featuring a mathematically rigorous and innovative three-step workflow that effectively tackles high-dimensional data challenges and overcomes the limitations of traditional Cox models. This framework enables integrated genomic and proteomic analysis, addressing a critical gap in single-omics approaches and providing a generalizable paradigm that better reflects biological complexity. Furthermore, the model demonstrates high persuasiveness through comprehensive multi-scenario simulations, consistently showcasing superior statistical power and robust false discovery rate control compared to other methods.

However, the study has several critical issues that require further refinement.

1. Validation of core model assumptions and clarification of applicability boundaries are insufficient: The outcome model of SMAHP is based on a parametric AFT model, assuming the error follows a normal distribution. However, the authors did not validate the validity of this distribution assumption in either simulated data or real-world data (CPTAC). If the actual data (e.g., survival time distribution) deviates from the normal assumption, it may lead to biased estimates of mediation effects. Additionally, the study did not investigate the applicability of the "proportional effect assumption" (linear associations between exposure and mediator, and between mediator and outcome). For example, it did not test for non-linear relationships between gene expression levels and protein abundances, nor did it explain the model’s robustness when interactions exist between variables. It is recommended to supplement residual distribution tests (e.g., Q-Q plots), non-linear relationship tests (e.g., adding quadratic terms), and clarify the limitations of the model when assumptions are violated.

2. Simulated data design is overly idealized and inconsistent with the characteristics of real-world data: In the simulation experiments, exposure variables (X) were set to follow a multivariate normal distribution with no correlations, and the association patterns of mediator variables (M) (e.g., 40% of M associated with 10% of X) were artificially predefined. This differs significantly from the characteristics of real multi-omics data (e.g., high co-expression among genes, heterogeneous associations between proteins and genes, and batch effects). Such an idealized design may overestimate SMAHP’s performance in real-world data. For example, if strong collinearity exists among exposure variables in real data, the variable screening performance of MCP-penalized regression may decline, but this scenario was not simulated. It is recommended to supplement simulation settings that better mimic real multi-omics data (e.g., introducing variable correlations, batch effects, and outliers) and re-validate SMAHP’s performance to ensure the method’s reliability in complex data scenarios.

3. Real-world data validation lacks depth and breadth: The current application is limited to single-cohort CPTAC head and neck squamous cell carcinoma (HNSCC) data (final sample size: 102 patients, after excluding 7 patients with missing overall survival (OS) data). External validation using multi-center, multi-disease data is lacking, making it difficult to demonstrate SMAHP’s generalizability. For example, the finding that "LCE3E mediates the effect of HMGB1P23 on OS" was not validated in other cancer types (e.g., lung cancer, breast cancer) or independent HNSCC cohorts, leaving open the possibility that this pathway is a cohort-specific occasional result. Furthermore, the application study did not extend the analysis to meet clinical practice needs—for instance, it did not explore the association between the mediation pathway and treatment response (e.g., chemotherapy sensitivity), nor did it validate the model’s performance in real-world data with small sample sizes or high censoring rates. It is recommended to supplement external validation with multi-cohort data (e.g., TCGA data, independent clinical study data), conduct analyses on multi-omics data from different disease types, and deepen result interpretation by integrating clinical indicators to enhance the method’s practical application value.

4. Robustness of causal inference assumptions is not fully demonstrated: The core assumption of causal mediation analysis—"no unmeasured confounding" (i.e., no omitted confounders affecting the associations between exposure and outcome, mediator and outcome, or exposure and mediator)—was not validated. The authors did not discuss potential unmeasured confounders in the CPTAC data (e.g., patient smoking history, alcohol consumption, comorbidities), nor did they perform sensitivity analyses (e.g., E-value calculation, simulating the effects of unmeasured confounders) to assess the impact of these factors on mediation effect estimates. Additionally, the study did not consider "interactions between mediators". In real biological systems, proteins often exhibit regulatory relationships, but SMAHP did not include such interaction terms, potentially missing key causal pathways. It is recommended to supplement sensitivity analyses to validate the rationality of the "no unmeasured confounding" assumption, or extend the analysis by incorporating mediator interaction terms into the model to improve the rigor of causal inference.

Minor Comments

1. It is recommended to supplement the simulation experiments with an analysis of "the impact of variable dimensionality (p/k) on computational time" to provide readers with references for computational cost when selecting the method.

2. Figure 1 (causal pathway diagram) could be supplemented with 95% confidence intervals to intuitively illustrate the uncertainty of effect estimates.

3. The Discussion section should explicitly propose optimization directions for SMAHP in "scenarios with explosive growth in multi-omics data volume" (e.g., integration of parallel computing, dimensionality reduction algorithms) to provide insights for future research.

Reviewer #3: The authors propose the SMAHP framework, which is a novel designed for high-dimensional causal mediation analysis integrating genomic, proteomic, and survival data. This addresses an important gap in existing methods which focus on single exposures, limited applicability to complex multi-omics data where multiple genes may work through multiple proteins to affect survival. The proposed three-step approach combining penalization, sure independence screening, and hypothesis testing is computationally feasible and demonstrates good performance in simulations under the conditions tested. The approach is technically sound, and the paper is generally well-written. However, the current manuscript would be strengthened by addressing several issues:

1. The authors indicate that the exposure variables (genes) are not correlated. By simulating independent genes, the authors test their method under a best-case scenario that likely overestimates performance. When genes are correlated, penalization methods like MCP would have to choose among correlated signals. This could lead to unstable variable selection and difficulty determining which gene is the true mediator versus simply correlated with it. Have the authors examined correlations among the top 100 genes selected from CPTAC data? If HMGB1P23 is highly correlated with other top-ranked genes, how confident can we be in the specificity of this finding?

2. The authors specifically select 100 genes and 200 proteins from pre-screening for SMAHP. Is the selection based on computational constraints or other considerations? Given that the starting dataset contains 60,669 genes and 9,469 proteins, reducing to 100 genes and 200 proteins seems to be a stringent threshold and could lose a true signal ranked just outside these cutoffs. Sensitivity analyses by repeating the analyses with alternative cutoffs could be helpful.

3. Based on my understanding, the method tests each protein mediator independently. However, proteins could have multiple weak sequential effects that combine to create a strong overall pathway. A gene can influence one protein and subsequently activates another protein, then affects survival. SMAHP tests each link separately and may find each individual effect too weak to reach significance even though the complete cascade has a substantial cumulative effect on survival outcome. While this limitation is shared by some other high-dimensional methods, the authors could further clarify this as a current limitation.

4. The authors mentioned that the proposed method identified a gene-protein pair associated with survival, specifically HMGB1P23 and LCE3E. To further elucidate the biological relevance of the identified HMGB1P23–LCE3E–survival pathway, the author could conduct additional analyses? For example, examining the correlation between HMGB1P23 gene expression and LCE3E protein abundance? Stratified survival analyses across joint expression strata of HMGB1P23 and LCE3E might also help to understand whether the mediation pattern translates into distinct survival outcomes. It would also be helpful to evaluate the robustness of the findings with other tests, which would strengthen confidence in the observed HMGB1P23–LCE3E mediation effect.

**Have the authors made all data and (if applicable) computational code underlying the findings in their manuscript fully available?**

The PLOS Data policy requires authors to make all data and code underlying the findings described in their manuscript fully available without restriction, with rare exception (please refer to the Data Availability Statement in the manuscript PDF file). The data and code should be provided as part of the manuscript or its supporting information, or deposited to a public repository. For example, in addition to summary statistics, the data points behind means, medians and variance measures should be available. If there are restrictions on publicly sharing data or code —e.g. participant privacy or use of data from a third party—those must be specified.requires authors to make all data and code underlying the findings described in their manuscript fully available without restriction, with rare exception (please refer to the Data Availability Statement in the manuscript PDF file). The data and code should be provided as part of the manuscript or its supporting information, or deposited to a public repository. For example, in addition to summary statistics, the data points behind means, medians and variance measures should be available. If there are restrictions on publicly sharing data or code —e.g. participant privacy or use of data from a third party—those must be specified.

Reviewer #1: Yes

Reviewer #2: None

Reviewer #3: None

PLOS authors have the option to publish the peer review history of their article (what does this mean?). If published, this will include your full peer review and any attached files.). If published, this will include your full peer review and any attached files.

.

Reviewer #1: No

Reviewer #2: **Yes:** Erping LongErping Long

Reviewer #3: No

**Figure resubmission:**
---

## [Editor Report · Decision Letter 1]

25 Mar 2026

PCOMPBIOL-D-25-01410R1

A multi-omics framework for survival mediation analysis of high-dimensional proteogenomic data

PLOS Computational Biology

Dear Dr. Ahn,

Thank you for submitting your manuscript to PLOS Computational Biology. After careful consideration, we feel that it has merit but does not fully meet PLOS Computational Biology's publication criteria as it currently stands. Therefore, we invite you to submit a revised version of the manuscript that addresses the points raised during the review process.

We look forward to receiving your revised manuscript.

Kind regards,

Chris Amos

Guest Editor

PLOS Computational Biology

Ferhat Ay

Section Editor

PLOS Computational Biology

**Editor Comments :**

The authors have responded well to the reviewers' comments and this is an important and very detailed manuscript. Therefore, I am suggesting accepting pending the minor revision that the manuscript must include within the body of the text a statement about the availability of code. The response to the review states ' The SMAHP (all upper cases) R package is freely available in the Comprehensive R Archive Network (CRAN) repository (https://cran.r-project.org/web/packages/SMAHP/index.html). Please reach

out to the corresponding author (Seungjun Ahn, seungjun.ahn@mountsinai.org) if you have any further inquiries about the R package.' but I do not see this comment included in the manuscript. Therefore, I do not know if this package would be accessible to the readers.

**Figure resubmission:**
---

## [Editor Report · Decision Letter 2]

9 Apr 2026

Dear Dr. Ahn,

We are pleased to inform you that your manuscript 'A multi-omics framework for survival mediation analysis of high-dimensional proteogenomic data' has been provisionally accepted for publication in PLOS Computational Biology.

Best regards,

Chris Amos

Guest Editor

PLOS Computational Biology

Ferhat Ay

Section Editor

PLOS Computational Biology

We just asked for some minor revisions to your paper. It seems like a very interesting and novel contribution

---

## [Editor Report · Acceptance letter]

PCOMPBIOL-D-25-01410R2

A multi-omics framework for survival mediation analysis of high-dimensional proteogenomic data

Dear Dr Ahn,

I am pleased to inform you that your manuscript has been formally accepted for publication in PLOS Computational Biology. Your manuscript is now with our production department and you will be notified of the publication date in due course.

With kind regards,

Judit Kozma
